# Towards Multimodal Deep Learning for Predicting Survival of Brain Metastatic Cancer

Inês N. Carvalho *[1], Carla Guerreiro[2], Cristiano Esteves[2], Matilde Calheira[1], Stefani Domentean[3], Rita Cascão[3], Mariana M. Santos[1], Cláudia C. Faria[3, 4], and Joao C. Guimaraes[1]

[1]Faculdade de Medicina, Universidade de Lisboa, 1649-028 Lisboa, Portugal
[2]Serviço de Imagiologia, Hospital de Santa Maria, Centro Hospitalar Universitário Lisboa Norte (CHULN), 1649-035 Lisboa, Portugal
[3]Gulbenkian Institute for Molecular Medicine (GIMM), Avenida Prof. Egas Moniz, 1649-035 Lisboa, Portugal
[4]Serviço de Neurocirurgia, Hospital de Santa Maria, Unidade Local de Saúde de Santa Maria (ULSSM), 1649-035 Lisboa, Portugal
`ines.c@edu.ulisboa.pt`

## Abstract

Brain metastases (BM) are associated with poor prognosis and high morbidity, yet survival prediction remains limited. Existing methods often rely on radiomic features, overlooking raw imaging and heterogeneous clinical data. We propose a deep learning (DL) framework that integrates multisequence magnetic resonance (MRI) and clinical variables for survival prediction. Our study compares a foundation model (FM) paired with a classifier against conventional convolutional neural networks (CNNs) to develop a robust model.

## 1 Introduction

Brain metastases are associated with a poor prognosis and high morbidity and mortality among cancer patients. Surgical intervention is often considered for selected patients, but it does not always improve quality of life or substantially prolong survival due to postoperative complications and the advanced nature of the disease [1, 2].

Deep learning (DL) has shown remarkable potential in radiology, enabling automated segmentation, diagnosis, and prediction of clinical outcomes [3]. Current studies on survival prediction for brain metastases have mainly focused on radiomic features, often omitting the direct use of imaging data [4, 5]. While several authors have highlighted the benefits of using multimodal data [6, 7], studies that directly integrate MRI images with more diverse clinical data—including previous treatments—and histological information remain limited [5].

To address these challenges, we leverage DL to integrate MRI sequences with clinical variables for survival prediction and individualized treatment guidance. We compare FMs, large neural networks trained on diverse datasets that perform well even with limited labeled data [8, 9], with traditional CNNs. By combining multisequence MRI and clinical information, our approach aims not only to improve prognostic reliability but also to systematically evaluate and compare these two methodological frameworks in developing a clinically relevant decision-support model.

## 2 Data

We are collecting a dataset of 288 patients from Santa Maria Hospital, Lisbon, diagnosed with brain metastases between [period 2009-2024]. MRI scans were acquired 1–3 days before brain surgery, and include T1, T1 post-contrast (T1*), T2, and FLAIR sequences for all patients, as well as additional imaging modalities such as Apparent Diffusion Coefficient maps, perfusion when available. Moreover, it contains detailed clinical information, including tumor histology, primary tumor characteristics, and comprehensive treatment records.

### 2.1 Imaging Preprocessing

MRI sequences are first converted from DICOM to NIfTI format using `dcm2niix` [10] and preprocessed following the pipeline (CBICA-CaPTk ). Briefly, MRI scans are first reoriented to LPS/RAI and then registered to the SRI24 atlas [11], including a temporary N4 bias field correction to compute the transformation matrix. Images are resampled to isotropic $1 \times 1 \times 1$ mm³ voxel spacing. Rigid registration is applied for T1, T2, and FLAIR to T1*, followed by rigid registration of T1* to the atlas. The resulting transformations are applied to all reoriented images, so that T1, T2, and FLAIR are mapped first to T1* space and then to atlas space. Finally, skull-stripping is performed using the HD-BET deep learning model [12]. For modeling, the MRI scans are stacked into 4D (4-channel) volumes in the order T1, T1*, T2, and FLAIR, which are then resized to $96 \times 96 \times 96$ voxels and intensity-normalized.

---

*Corresponding Author.

# 3    Preliminary Experiments

We first conducted an analysis of our target variable, survival, and framed the survival prediction task as a classification problem, taking into account both target variable distribution and clinical usability in the context of surgical decision-making. We examined the distribution of the target variable, checked for class balance, and evaluated correlations with follow-up time and mortality.

After applying the necessary preprocessing steps described previously, and with the goal of comparing the performance of FM and CNNs, we selected a Vision Transformer-based autoencoder implemented in PyTorch and MONAI. Specifically, we employed the `ViTClassifier_block16` (ViT-16, with a dropout block size of $16 \times 16 \times 16$). This model leverages pretrained weights from large-scale medical imaging datasets [13]. Latent representations were computed during the forward pass for 151 patients[1]. Each MRI sequence was initially evaluated independently, with the remaining sequences set to `NULL`, and missing latent variables treated as constants in the UMAP representation. Also, latent representations were computed considering the complete 4D volumes.

# 4    Preliminary Results

## 4.1    Survival as a Categorical Variable

To ensure clinically meaningful and balanced data, we stratified survival into three categories: <6 months, 6–21 months, and >21 months. This resulted in 21.87%, 26.39%, and 51.7% of patients in the short, medium, and long survival groups, respectively, with only 10 patients having medium survival and a follow-up shorter than 21 months (Figure 1).

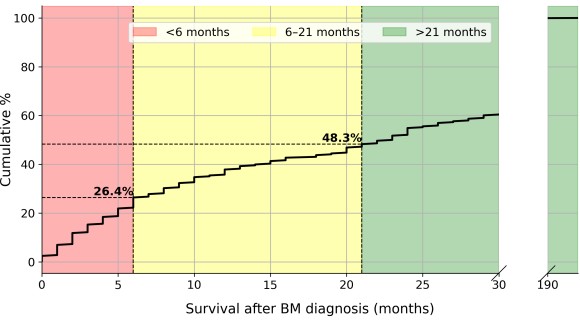

**Figure 1.** Cumulative distribution function (CDF) of survival after BM diagnosis, highlighting the three survival categories.

## 4.2    FM - Latent Embeddings

After processing the MRI sequences, as previously described, we computed and then examined how the latent space distributions relate to the categorical survival variable, as visualized using UMAP (Figure 2). No clear separation was observed in the UMAP projection of the latent space across MRI sequences (Figure 2(b)). Although some local distinctions are visible, the latent representations do not exhibit a consistent or meaningful relationship with the predefined survival categories.

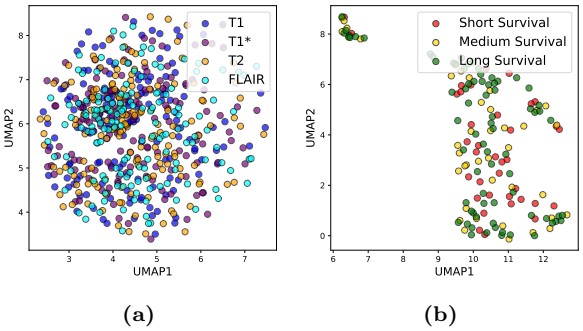

**(a)**                    **(b)**

**Figure 2.** UMAP projection of latent representations from the pretrained ViT-16. (A) Colored by MRI sequence. (B) Colored by categorical survival groups.

# 5    Future Experiments

The current results suggest that the pre-trained FM embeddings do not accurately distinguish the defined categorical targets. Fine-tuning these embeddings, along with adapting the simple classification architecture proposed [13]—especially the average pooling and FC layers may be needed.

Future experiments will use our dataset with an 80/10/10 split for training, validation, and testing, along with external validation on a public dataset [14].

We will compare the FM with several architectures available in MONAI, including DenseNet121, ResNet18, and SENet154 (previously used by the authors [13]), as well as ResNet34/50/101, DenseNet169/201, and EfficientNet, applying Med3D transfer learning [15] where appropriate.

Our primary focus will be survival prediction, while also exploring outcomes such as primary tumor origin, BM relapse, molecular subtype, and markers, either via multi-output models or separate models per task.

Finally, we will develop an integrated multimodal framework combining MRI with clinical and histology data to enhance predictive performance and enable comprehensive multimodal analysis.

---

[1] Of 288 patients, 151 had complete MRI data for analysis.

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
