# OpenReview forum: "Towards Multimodal Deep Learning for Predicting Survival of Brain Metastatic Cancer"
_NLDL.org/2026/Abstracts_Track — NLDL 2026 Abstracts_

### Official Review · Reviewer_qtBE · 2025-10-30

**Soundness:** 3
**Correctness:** 2
**Rating:** 4
**Confidence:** 5

**Summary:**

The authors test a multimodal pipeline that integrates different MRI modalities with clinical variables for survival prediction on a private in-house dataset of metastatic brain cancer patients.
The authors compare the performance of a foundation model paired with a classifier to a standard CNN.

**Strengths:**

The abstract covers a relevant topic for the medical image community, namely the integration of multiple modalities.
Per se, the authors do not propose any technical novelty, however the application to a curated in-house dataset with different clinical data types is of significant interest.
The figures are nice and the image processing pipeline is described in detail.

**Weaknesses:**

The abstract presents a few weaknesses in terms of clarity and presentation of results.

In the abstract, the authors claim that they integrate different imaging modalities with other clinical variables.
However, in Sections 2.1–4, every described preprocessing and analysis step concerns MRI preprocessing (registration, normalisation, 4D stacking, etc.) and foundation model latent representations, with no reference to clinical data integration.
In Section 5, they mention plans to “develop an integrated multimodal framework combining MRI with clinical and histology data” — implying this integration is a future goal, not something currently done.
Therefore, the multimodal claim in the abstract is premature and unsupported by the presented methods or results. The current study uses only MRI data, and the integration with clinical variables is merely proposed for future work.

The authors state that they compare a foundation model with CNNs, however no details are provided about the used CNN; additionally, no results are provided for the CNN.
So at this stage, despite the claim in the abstract, no CNN comparison results are actually presented, the manuscript reports only preliminary FM analysis. The stated comparison is not yet supported by data, and that CNN results (or at least a description of planned comparison metrics) should be included to justify the claim.

In the abstract, the authors claim that their pipeline consists of a foundation model combined with a classifier. However, the only results presented consist of an unsupervised analysis (UMAP) of the features extracted by the foundation model. No results for the linear probing are provided, which is in contrast with the initial claim.

I think it would be interesting to present this work at the conference and the authors could get some constructive feedback for the future steps in this preliminary work. For the poster presentation, I would recommend the authors to switch the focus on the analysis of the representations of the foundation model or include some of the missing elements mentioned above.

---

### Official Review · Reviewer_S74x · 2025-10-31

**Soundness:** 3
**Correctness:** 3
**Rating:** 4
**Confidence:** 4

**Summary:**

The paper introduces an extensive dataset focused on brain metastases that integrates multiple clinical and imaging modalities and provides a well-documented preprocessing pipeline for the imaging data. The resource is positioned to enable multimodal learning and downstream clinical prediction tasks, including survival analysis.

**Strengths:**

- New multimodal dataset for brain metastasis: The collection spans MRI, apparent diffusion coefficients (ADC), perfusion measures, tumor histology, primary tumor characteristics, and patient treatment records. This breadth is clinically valuable and comparatively rare, positioning the dataset as a strong foundation for cross-modality research.

- Detailed image preprocessing pipeline: The manuscript provides a thorough description of the imaging preprocessing, which improves transparency, facilitates reproducibility, and should reduce variance introduced by site- or scanner-specific quirks.

**Weaknesses:**

- Multimodality not yet demonstrated: Although the dataset is inherently multimodal, the current work does not yet conduct multimodal modeling or present fusion baselines.
- Foundation model unspecified: The paper does not state which foundation model was used, hindering reproducibility and making it difficult to contextualize the reported embeddings or model behavior.
- Survival binning rationale: The manuscript does not justify the chosen categorization scheme for survival outcomes. Without a clinical or methodological rationale—and without addressing censoring—this may reduce statistical efficiency and obscure clinically meaningful differences.
- Lack of CNN comparisons: Comparisons against standard CNN-based imaging baselines are not yet included, leaving unclear how classical approaches perform relative to the proposed setup.

Minor Issues:
- Line 123 wording: Please clarify that the model, not the embeddings, is fine-tuned.

---

### Official Review · Reviewer_c9Us · 2025-11-03

**Soundness:** 2
**Correctness:** 3
**Rating:** 2
**Confidence:** 2

**Summary:**

The paper considers the use of a foundation model and a CNN to predict survival in patients with brain metastatic cancer, based on MRIs together with other relevant features such as treatment history. Several MRIs per patient are used, and their preprocessing is described in detail. They train a network and plots the latent features, but no clear clusters can be found. They suggest future work to improve the method, as well as to extend it for other classification tasks in this setting and as a desicion support tool.

**Strengths:**

Incorporating MRI scans with other medical data is an interesting challenge. Extensive suggestions for future work. Many implementation details are clearly communicated.

**Weaknesses:**

It is not well motivated why predicting survival is useful, nor how this could be extended to decision support systems. No evaluation is performed on actual prediction capabilities. The purpose of Figure 1 and the meaning of the cumulative y-axis is unclear. It is not well motivated why the categories for survival is chosen as is, and the method is not able to distinguish between the categories in latent space. It is unclear how the additional medical information is incorporated in the model, which seems to be the main contribution of the paper and thus a very important aspect to report.

---

### Decision · Program_Chairs · 2025-11-05

**Decision:**

Accept

**Comment:**

The reviewers found the abstract borderline, yet the PCs believe it will be of interest to the community and should have the opportunity be presented.